# Cost-effectiveness of Zvandiri, a community-based support intervention to reduce virological failure in adolescents living with HIV in Zimbabwe: Results of a decision analytical model

Collin Mangenah[1,2]*, Webster Mavhu[1,2], Nicola Willis[3], Juliet Mufuka[1],
Sarah Bernays[4], Walter Mangezi[5], Tsitsi Apollo[6], Ricardo Araya[7],
Helen A. Weiss[8], Alfredo Palacios[9], Fern Terris-Prestholt[10,11], Frances M. Cowan[1,2],
Hendramoorthy Maheswaran[12]

**1** Centre for Sexual Health and HIV/AIDS Research (CeSHHAR), Harare, Zimbabwe, **2** Department of International Public Health, Liverpool School of Tropical Medicine, Liverpool, United Kingdom, **3** Zvandiri, Harare, Zimbabwe, **4** School of Public Health, University of Sydney, Sydney, Australia, **5** Department of Psychiatry, University of Zimbabwe College of Health Sciences, Harare, Zimbabwe, **6** AIDS and TB Unit, Ministry of Health and Child Care, Harare, Zimbabwe, **7** Health Services and Population Research Department, King's College London, London, United Kingdom, **8** MRC International Statistics and Epidemiology Group, London School of Hygiene & Tropical Medicine, London, United Kingdom, **9** Centre for Health Economics, University of York, York, United Kingdom, **10** Warwick Centre for Global Health, Warwick University, Coventry, United Kingdom, **11** Policy, Advocacy and Knowledge Branch, UNAIDS, Geneva, Switzerland, **12** Imperial College London, London, United Kingdom

\* cmangenah1@gmail.com, collin.mangenah@ceshhar.org, colin.mangenah@lstmed.ac.uk

## Abstract

Improving antiretroviral therapy (ART) adherence among adolescents living with HIV (ALHIV) improves outcomes, but with resource implications. We conducted a cost-effectiveness analysis extrapolating the costs and benefits of a community-based peer-support intervention (Zvandiri) among ALHIV in Zimbabwe. We used a de-novo multistate Markov decision-analytic model that simulated Zvandiri lifetime costs and benefits on viral suppression, death rates, life-years (LY) and quality-adjusted-life-years (QALYs) gained from the healthcare system perspective. We estimate the incremental cost-effectiveness ratio (ICER) per LY and QALY gained and compare the ICER to proposed cost-effectiveness thresholds of $500 and $700 per LY or QALY gained. We explore parameter uncertainty using probabilistic sensitivity analyses. Cohort-microsimulation suggests that after 40 years under SoC, 21% of 280 ALHIV will have undetectable viral-load (VL), 12% will have low VL (<1000 copies/mL), 10% will have high VL (≥1000 copies/mL) and 57% would have died. With Zvandiri, ART adherence improves, decreasing annual probability of virological failure or death. After 40 years, 65% will have undetectable viral load, 23% low VL, 3% high VL and 9% would have died. Zvandiri results in 1,345 LYs gained at incremental cost of $500,587, yielding a discounted ICER of $372 per LY gained. Zvandiri also results in 1,246 QALYs at incremental cost of $123,645, yielding a discounted ICER of $99

**Data availability statement:** A spreadsheet (Zvandiri Markov Model Parameters) with all model data pertinent to the cost effectiveness analysis has been submitted together with the manuscript and other supporting materials. Potentially sensitive data for the cohort of individual adolescents living with HIV used to estimate the transition probabilities is held and available upon reasonable request via a non-author point of contact: Institution: Centre for Sexual Health and HIV/AIDS Research (CeSHHAR) Contact person: Data Manager, Jeffrey Dirawo (Jeffrey.dirawo@ceshhar.org).

**Funding:** The research reported in this publication was funded by ViiV Healthcare's Positive Action for Adolescents Programme (Grant # 1710223/OV192217 to FMC). The funders had no role in study design, data collection and analysis, decision to publish, or preparation of the manuscript. The content is solely the responsibility of the authors and does not necessarily represent the official views of ViiV Healthcare.

**Competing interests:** The authors have declared that no competing interests exist.

per QALY. The ICER is highly sensitive to programme costs, health-related utilities, and the discount rate. Zvandiri is a cost-effective intervention for reducing virological failure and death in ALHIV. Our analysis likely underestimates the full benefits of the intervention by not accounting for reductions in HIV transmissions resulting from higher virological suppression observed in full transmission models.

## Introduction

As the proportion of adolescents living with HIV (ALHIV, aged 10–19) continues to increase globally, largely because more perinatally infected children are surviving into adolescence, a significant concern arises from their higher attrition rates from HIV treatment and care compared to children and adults, leading to greater instances of treatment failure, morbidity and mortality [1–6]. This is especially noticeable in sub-Saharan Africa (SSA), which is home to 84% of the world's ALHIV (still projected to reach 50% by 2050) and where adolescents are the only demographic experiencing rising mortality rates in this region [5,7].

Scalable, effective and cost-effective strategies for differentiated support maximising not only wellbeing but adherence to treatment and care are needed [8]. Primary care facilities however frequently lack adequate capacity to provide effective support for adherence as well as the broader psychosocial issues that children and adolescents grapple with [8]. Several analyses have explored the effectiveness of community-based support programmes which the World Health Organisation recommends for improving the wellbeing and longevity of ALHIV in Eastern and Southern Africa [1,5,6,8]. Policymaking decisions regarding adoption of public health interventions must however consider not only effectiveness but value for money particularly given the current international HIV funding crisis [9].

Promising public health interventions urgently need cost-effectiveness analyses to help inform policymakers and health providers on how best to allocate scarce resources [10]. In particular, the evidence on cost-effectiveness of comprehensive health systems strengthening programmes that improve retention in care is lacking, particularly for ALHIV, yet evidence is critical for guiding scaleup in SSA countries [11–13]. A study in East Africa evaluated the cost-effectiveness of the most effective retention-in-care strategies and to contrast them with alternative resource allocation options for individuals living with HIV, particularly in relation to expansion of access to ART [14]. The study findings suggest possibility of scaling up retention programmes at relatively low cost, which would serve as a valuable addition to the options available for policy and decision-makers in the East African region [14].

Zimbabwe, one of the countries hardest hit by HIV (adult prevalence was 10.49% in 2023) and with 1.3 million people living with HIV in 2023, revised its national ART guidance in December 2013, in line with the 2013 WHO consolidated ART prevention and treatment guidelines, which recommended innovative models of ART delivery (decentralisation, integration and task-shifting) [15–17]. The 2013 Zimbabwe ART guidelines sought to simplify recommended treatment regimens for children and

adolescents to facilitate decentralisation of paediatric HIV care to primary healthcare facilities with the aim of improving access and coverage [18,19]. Currently, in line with the WHO 'Test and Treat' ('treat all'), and to ensure improved survival and quality of life of persons living with HIV as well as reducing chances of HIV transmission, Zimbabwe national policy and guidelines (2016) recommend immediate ART initiation for all irrespective of disease stage or CD4 count [20–22]. By 2016, overall ART coverage was 68% (83% in the 0–14 age group and 66% in the adult population) [11].

Zvandiri ('As I am') (www.zvandiri.org), a theoretically grounded 'WHO best practice' programme is described as "a peer-led, multi-component differentiated service-delivery model that aims to directly improve the health and wellbeing and strengthen engagement of children, adolescents, and young people living with HIV with services across the HIV prevention and care cascades" [23]. Delivered by trained, mentored peer counsellors known as CATS (community adolescent treatment supporters), aged 18–24, it aims to improve HIV diagnosis, disclosure, linkages, adherence, retention and provide ongoing support for young people's mental health, social protection and, sexual and reproductive health [8,23–28]. The effectiveness and economic costs of the Zvandiri intervention on HIV clinical and psychosocial outcomes among ALHIV in Zimbabwe were evaluated in a cluster-randomised trial (CRT) conducted between 2016 and 2018 [8]. The intervention was effective at increasing rates of viral suppression over 96 weeks and in the trial-based economic analysis, we found the annual costs per patient on ART and per viral suppression were quite high at $997 and $1,340, respectively [8]. However, Zvandiri could have the potential to yield health benefits beyond the trial time horizon.

We simulate the cost-effectiveness and potential impact of the enhanced adherence intervention (Zvandiri) on viral suppression, death rates, life years (LYs) gained, quality adjusted life years (QALYs) and incremental cost-effectiveness ratios (ICER) compared with standard of care (SoC) over a 40-year horizon. In this study, we assess whether the incremental costs of the Zvandiri intervention yield sufficient benefits in LYs to be cost-effective over a longer time horizon. We also model QALYs gained as an outcome measure integrating gains in an individual's health-related quality of life from reduced morbidity and mortality [29–33].

## Materials and methods

### Study design

We used a de-novo multistate Markov cohort simulation model to extrapolate lifetime costs, LYs and QALYs 40 years beyond the trial (2019–2058) [8,31,34–36]. The model was informed by ART utilisation service data and related full economic costs measured and valued in a prospective primary costing study conducted alongside the Zvandiri CRT from which QALY data were also generated. Full economic costs here refer to both intervention delivery and direct ART costs. The QALY is a generic measure of disease burden, including both the quality and quantity of life lived, with one QALY equating to one year in perfect health [37]. QALY scores range from 1 (perfect health) to 0 (dead) [29,38]. Hybrid economic models incorporating both within-trial and model-based (beyond-trial) methodologies take advantage of the in-trial data to inform the modelling effort, address the limitations of short-term (truncated) follow-up within trials, and project the results through time, generating a range of plausible projections of longer-term outcomes [38,39].

### Zvandiri trial overview

Both the Zvandiri trial (2016–2018) and primary costing design have been described in detail previously [8]. In brief, sixteen clinics were randomised 1:1 to Ministry of Health (MoH) SoC or MoH SoC augmented by the Zvandiri intervention (enhanced and differentiated support). The trial was located in two districts (Bindura and Shamva) in Mashonaland Central province of Zimbabwe where ART coverage among ALHIV was lowest nationally (29%) at the start of the trial [8]. The evidence informing parameter estimates, comprising the state transition probabilities (between viral suppression, virological failure and deaths), economic costs and health-related utilities were primarily derived from the trial, which enrolled 212 and 284 adolescents in the Zvandiri and SoC groups, respectively. These data were accessed for the purposes of

decision analytical modelling between 10 October 2018 and 20 February 2019. Authors had no access to information that could identify individual participants during or after data collection as all data were de-identified, using only unique participant identifiers.

## Standard of care

For participants in the control group, adherence support was provided by adult counsellors and nursing staff during clinic visits. Following ART initiation, ALHIV were seen three monthly, with viral load monitoring conducted at six monthly intervals. Prescription refills, pill counts and consultations with clinical staff were used to measure adherence [8].

## The intervention

Participants in the intervention group received MoH SoC (ART and adherence support provided by an adult counsellor) supplemented by the Zvandiri intervention. Trial participant allocation to a designated CATS depended on residential area. CATS were integrated within Government health facilities and allocated a caseload of ALHIV to counsel and support through monthly support group meetings, monthly home visits, daily or weekly text message reminders, weekly phone calls, health centre contact and caregiver workshops. At each clinic, CATS were supervised by a designated clinic nurse, with additional support from a district Zvandiri mentor through weekly supervision meetings and WhatsApp [8].

The level of support was differentiated (i.e., standard or enhanced) according to the clinical and psychosocial needs of individual clients. 'Standard of care' was offered to adolescents with VL < 1000 copies/mL or a most recent (and within the past 6 months) CD4 count of at least 200 cells/mL or both, and 100% scheduled clinic visit attendance in past 3 months. This involved a home visit once a month, plus a weekly, individualised text message. 'Enhanced care' was offered to adolescents with ≥1000 copies/mL or CD4 count of less than 200 cells/mL or both, at risk of common mental disorders or a major depressive disorder, < 100% scheduled clinic visits in the past 3 months, had started ART in the past 3 months, who were pregnant, or had other psychosocial challenges or protection issues [8].

## Primary cost analysis

**Unit costs of ART treatment.** The primary costing approach has been previously described [8]. It followed recommendations contained in international costing guidelines taking the Zimbabwe MoH perspective [35,36]. We estimated the full annual economic costs (January to December 2018) for 16 government health facilities which provide HIV treatment to all age groups, with eight additionally providing the Zvandiri intervention. All costs were calculated in 2018 United States Dollars (US$) using the prevailing 1:1 exchange rate at the time of data collection (Zimbabwe uses the US$ as its official currency since 2009). Estimating the cost per adolescent virally suppressed on ART required dividing the total programme cost by the total number of adolescents known to be virally suppressed in each arm. Analyses were conducted in Stata version 13.1 (College Station, TX, USA), and Microsoft Excel [40].

**Trial outcomes.** *Clinical outcomes.* In the decision-analytic model, to appropriately assign QALYs to all patients who are alive, we modified the primary outcome by separating the proportion of participants who had virological failure from those who had died (and excluding non-HIV related deaths) as they could still transition back to states A (HIV viral load undetected) and B (HIV viral load <1000 copies/mL) (Fig 1). In the Zvandiri Trial, the primary outcome had been the proportion of participants who had died (of any cause) or had virological failure (VL > 1000 copies/mL at 96 weeks) [8]. At endline (96 weeks), virological failure or death was less common among participants in the Zvandiri group than in the SoC group (adjusted PR 0·58, 95% CI 0·36–0·94; p = 0·03).

**Primary unit costs of ART treatment.** Resource consumption during clinic visits was assessed and valued based on the 955 adolescent clients on ART in the 16 government clinics at the end of the trial period (432 in the Zvandiri arm; 523 in the control arm) [8]. Based on the primary trial costing (discussed above in Unit costs of ART treatment) data, in the SoC arm, routine care (ART) costs (excluding above-site level non-governmental organisation (NGO) overheads, which

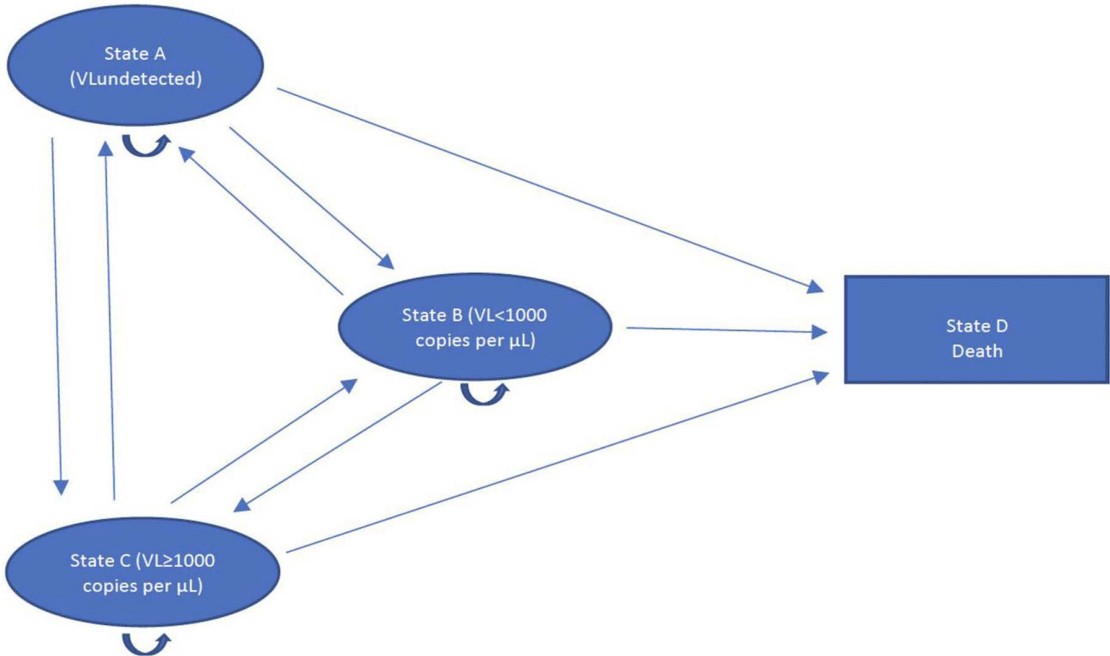

**Fig 1. Four-state Markov cohort-simulation model structure used to estimate the cost-effectiveness of a community-based support intervention to reduce virological failure in adolescents living with HIV in Zimbabwe (Zvandiri).**

would not be incurred in a government programme) are assumed to be $166 and cost per virally suppressed adolescent is $256 annually. For the intervention arm, ART plus Zvandiri costs are US$324 and costs per adolescent virally suppressed on ART (including Zvandiri costs) are US$449, respectively.

## Cost-effectiveness modelling

**Study perspective.** We conducted a cost-effectiveness analysis from the Zimbabwe MoH perspective guided by international reporting guidelines for economic evaluations [32,35].

**Comparators.** We extrapolated the longer-term ("in perpetuity") costs and economic consequences of the Zvandiri trial that compared enhanced care versus routine ART and adherence support ("status quo scenario") in Zimbabwe over 40 years [31,34,41].

**Markov model.** A Markov multi-state model is a type of cohort simulation modelling framework that breaks diseases into a finite set of mutually exclusive and exhaustive health states, divides a simulated cohort of patients amongst these states, and simulates their disease by allowing transitions between states to represent all possible consequences of an intervention of interest over time [31,34,42–45]. Each individual represented can only be in one of the disease states at any given time and only moves (transitions) as their condition changes over discrete time periods (cycles).

Cycles can have varying durations of weeks, months or years. 'Transition probabilities' represent movements from one disease state to another (from one cycle into the next), with a cost and health outcome associated with time spent in each disease state per cycle [34]. Given the lifetime nature of ART and adolescent patients' ability to "transition" between states (moving from undetected VL levels to detectable or vice versa and back over time), the Markov model was considered the model of choice (Fig 1) [45].

Other approaches commonly used to conduct cost-effectiveness modelling are decision trees and microsimulation or individual sampling models [31,45,46]. Though, the simplest form of decision-analytical model "decision trees" are limited to single, once-off events and struggle to model complex, long-term scenarios with repeated transitions between health states, which is common in chronic diseases. Moreover, they may suffer from rapidly becoming unwieldy or impossible to construct over multiple cycles, creating overly complex trees (branchy) that do not generalise well (overfitting), instability, and poor extrapolation [45,46]. A key strength of microsimulation approaches is their ability to follow individuals rather than cohorts, incorporating memory and allowing a simulated individual's transitions to depend not only on the current state, but also on their past experiences and states. In the process, they create a "memory" within the model allowing for more realistic and complex behaviour simulations over time [31].

We began our model using a decision tree to determine feasibility and gather baseline data although, as discussed above, it is better suited for more straight forward, short-term problems [47]. We then used Markov modelling, although weakened by the "memoryless" assumption, instead of the more ideal but complex microsimulation approach (individual sampling) informed by the need for a simple and more parsimonious model, allowing us greater accuracy in our forecast and since we were less concerned with ALHIV's prior histories, but more with how their present state influenced future transitions [45].

## Model inputs

**Cohort characteristics.** Based on trial data, we modelled a population of adolescents (13–19 years) who were engaged in care and on prescribed ART [8]. Median age at model start was 15 years (IQR 14–17), 52% of adolescents were female, 81% were orphans, and 47% had a viral load of ≥1000 copies/mL.

Fig 1 is a four-state Markov cohort-simulation model structure used to simulate Zvandiri's cost-effectiveness. Lines between the four states represent transition probabilities, or the rates (likelihood) at which individuals in one health state "transition" to any of the other health states. Transition probability refers to the conditional likelihood of the individual staying in the same state or transferring to another during the cycle. In the model, states C (HIV viral load ≥1000 copies/mL) and D (only deaths due to HIV) are treated as separate outcomes, to facilitate utility assignment to patients who are failing, but remain alive (defined as a composite measure in the primary CRT) [8]. Likewise, states A (HIV viral load undetected) and B (HIV viral load <1000 copies/mL) are also treated as separate outcomes.

**Time horizon.** The costs and economic consequences of the Zvandiri trial were evaluated over a 40-year time horizon [41,48]. The additional 40 years were assumed to represent a lifetime horizon and considered long enough to capture or account fully for all expected and unexpected outcomes (lifetime costs and effects) of ALHIV on HIV treatment. The 40-year time horizon was also informed by the life expectancy at birth in Zimbabwe (in years) which, by 2019, had improved to 60.7 years, from 46.6 years in 2000 [49]. The global estimate during the same period was 73.3 years [49].

**Discounting.** To adjust for the fact that individuals typically place less value on future economic costs and outcomes than present ones (time value of money), we applied a standard constant (uniform) annual discount rate of 3.5% per annum on both costs and economic consequences [50–54].

**Natural history and treatment.** The cohort (280 ALHIV), registered for HIV care at one of the trial intervention clinics, aged 13–19 years, either starting or already on ART (hence diagnosed), enter the model (decision node) and experience yearly (annual) probabilities of clinical states (Table 3) or events (transition probabilities). For the entry cycle (year 0 = beginning at Zvandiri trial endline (96 weeks)), the entire cohort are assigned a state based on baseline viral load count, informed by actual changes observed in ALHIV health states in the CRT between baseline and 96 weeks: 1) HIV viral load undetected, 2) HIV viral load <1000 copies/mL, and 3) HIV viral load ≥1000 copies/mL. At 96 weeks (end of trial period) the presence or absence of effective ART and adherence support (function of individual ALHIV's levels of adherence) leads to transition of individual ALHIV from baseline status (baseline survey results) and between the

four states informing the model start simulations. Fig 2 is a decision tree depicting the start simulations in cycle 0 using transition probabilities based on trial endline at 96 weeks.

**Markov process and chain.** In general, Markov models are iterated at each cycle (or sequence) in two possible ways, a Markov process or Markov chain. Whereas the transition probabilities in a Markov process are dynamic (changing with each cycle) to capture for example, the likelihood in some diseases, of increased risk of dying with age, in this model we used a Markov chain where the transition probabilities are static (not changing with each cycle). For each successive cycle (representing 1 year) in the 40-year Zvandiri cohort-simulation model horizon, ALHIV move between the states according to the transition probabilities calculated based on the proportion of ALHIV moving between the three states and death (absorbing state) (Table 1).

Table 2 presents additional input parameters including relevant life year weights and costs assigned to each mutually exclusive health state experienced by individuals. At the end of the simulation, the model tallies clinical events (states), duration spent in each health state, life expectancies and, cost per ALHIV on ART.

**Health state valuations.** Health state valuations for LYs gained were determined by adding the total number of ALHIV in states A (HIV viral load undetected), B (HIV viral load <1000 copies/mL) and C (HIV viral load >1000 copies/mL). QALYs states A (HIV viral load undetected) and B (HIV viral load <1000 copies/mL) were assigned utilities=0.84 and 0.86 for SoC and Zvandiri, respectively based on trial endline data [8]. State C (viral load ≥1000 copies/mL) and the absorbing state D (death) were assigned utilities=0.3 and 0.0 respectively [33].

*Lifetable half-cycle correction.* We used the Barendregt lifetable half-cycle correction method to allow ALHIV patient events or transitions to occur randomly throughout each cycle rather than at the beginning or end, which may lead to

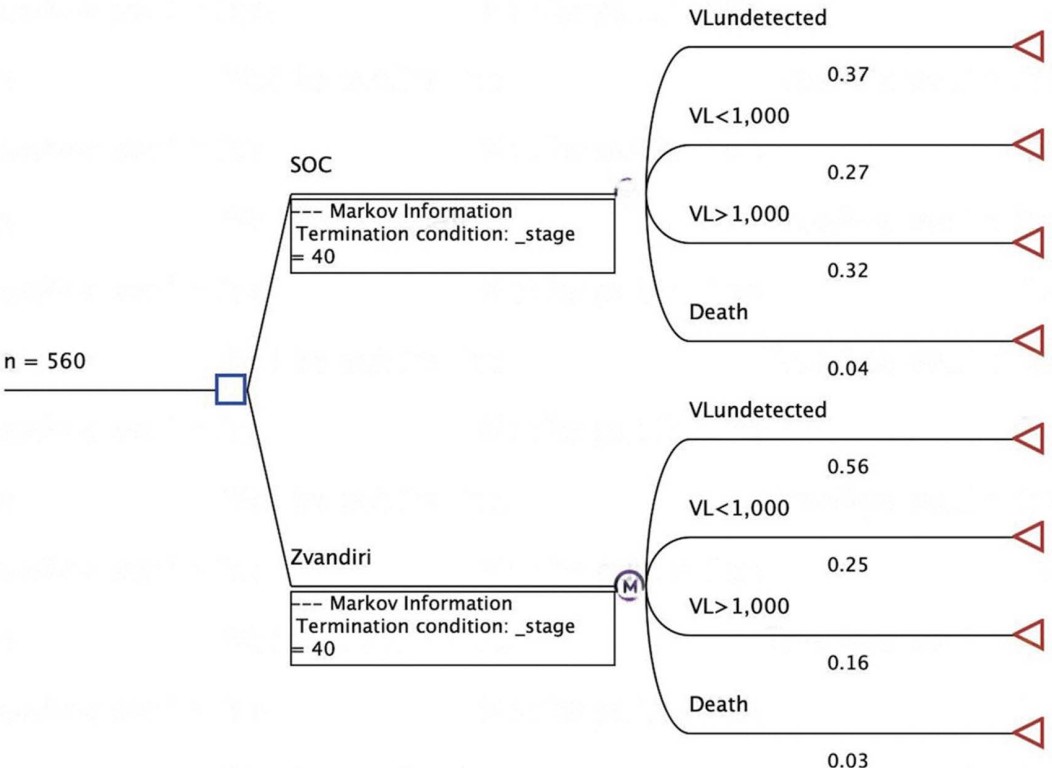

**Fig 2. Decision tree showing allocation of adolescent cohort at 96 weeks (beginning of model).**

PLOS Global Public Health

**Table 1. Transition probabilities for proportions of ALHIV moving between the three states and death.**

| Transition states | Variable | Probability | Probability (Zvandiri) | Description |
|---|---|---|---|---|
| Undetected -->Undetected | 1 - tp(Undetected_VL<1,000) - tp(Undetected_VL>1,000) - tp(Undetected_death) | 0.67 | 0.78 | Remain in the undetected state |
| Undetected -->VL<1,000 | tp(Undetected_VL<1,000) | 0.20 | 0.22 | Transition from undetected to VL<1,000 state |
| Undetected -->VL>1,000 | tp(Undetected_VL>1,000) | 0.12 | 0.00 | Transition from undetected to VL>1,000 state |
| Undetected -->death | tp(Undetected_death) | 0.00 | 0.00 | Transition from undetected to dead state |
| VL<1,000 -->VL undetected | 1 - tp(VL 1,000<1,000_VL<1,000) - tp(VL<1,000_VL>1,000) - tp(VL<1,000_death) | 0.42 | 0.56 | Transition from VL<1,000 to undetected state |
| VL<1,000 -->VL<1,000 | tp(VL<1,000_VL<1,000) | 0.43 | 0.35 | Remain in VL<1,000 state |
| VL<1,000 -->VL>1,000 | tp(VL<1,000_VL>1,000) | 0.13 | 0.09 | Transition from VL<1,000 to VL>1,000 state |
| VL<1,000 -->death | tp(VL<1,000_death) | 0.01 | 0.00 | Transition from VL<1,000 to death state |
| VL>1,000 -->VL undetected | 1 - tp(VL>1,000_VL<1,000) - tp(VL>1,000 _VL>1,000) - tp(VL>1,000_death) | 0.13 | 0.43 | Transition from VL>1,000 to VL undetected state |
| VL>1,000 -->VL<1,000 | tp(VL>1,000_VL<1,000) | 0.24 | 0.20 | Transition from VL>1,000 to VL<1,000 state |
| VL>1,000 -->VL>1,000 | tp(VL>1,000 _VL>1,000) | 0.55 | 0.30 | Remain in VL>1,000 state |
| VL>1,000 -->death | tp(VL>1,000 _death) | 0.08 | 0.07 | Transition from VL>1,000 to dead state |
| Dead _->Dead | tp(dead) | 1 | 1 | Remain in dead state (absorbing state) |

over- or underestimation of outcomes (LYs and QALYs gained) and the associated costs [55,56]. Lifetable half-cycle correction assumes that events occur at the mid-point of the cycle since we do not necessarily know at what point during each of the yearly cycles patients transition between states [55,57].

**Incremental cost-effectiveness ratio.** The incremental cost-effectiveness ratio (ICER) for Zvandiri over SoC was expressed as:

$$ICER = \frac{\text{Cost of Zvandiri intervention - Cost of SoC}}{\text{Effectiveness of Zvandiri intervention - Effectiveness of SoC}}$$

**Cost-effectiveness thresholds.** In this analysis, we compared the ICER to two commonly accepted thresholds, <$700 and <500 per LY or QALY gained, now generally considered more appropriate for low-and middle-income countries (LMICs) [58–60].

**Sensitivity and scenario analyses.** Parametric uncertainty refers to variation in the numerical input values of a decision analytic model. These parameters may include transition rates between health states, values governing the efficacy of interventions, as well as cost and utility weights associated with health states. Parametric uncertainty occurs when parameters are estimated from the results of epidemiological and economic studies that are subject both to sampling error and lack of validity due to differences in modelled and actual study populations. Probabilistic sensitivity analysis is often considered the best method of accounting for uncertainty in the joint distribution of the parameters [31,32,35,61].

Zvandiri has evolved from an NGO-led model with strong community-clinic linkages into a government-led national, decentralised approach which is coordinated by the National AIDS Council through district mentors (former CATS). This

**Table 2. Four-state Markov model parameter inputs.**

| | Standard of care | | | | Zvandiri | | | |
|---|---|---|---|---|---|---|---|---|
| Starting cohort | 280 | | | | 280 | | | |
| Trial horizon | 40 years | | | | 40 years | | | |
| State transitions | Undetected | VL<1000 | VL>1000 | Death | Unde-tected | VL<1000 | VL>1000 | Death |
| Undetected | 67% (59/86) | 20% (17/86) | 12% (10/86) | 0% (0/86) | 78% (42/54) | 22% (12/54) | 0% (0/54) | 0% (0/54) |
| VL<1000 | 42% (28/67) | 43% (29/67) | 13% (9/67) | 1% (1/67) | 56% (28/55) | 35% (19/55) | 9% (5/55) | 0% (0/55) |
| VL>1000 | 13% (16/127) | 24% (31/127) | 55% (70/127) | 8% (10/127) | 43% (40/98) | 20% (20/98) | 30% (29/98) | 7% (10/98) |
| State descriptions | Costs ($) | Utility weights | Annual discount rate - costs (%) | Annual dis-count rate - benefits (%) | Costs ($) | Utility weights | Annual discount rate - costs (%) | Annual dis-count rate - benefits (%) |
| Direct medical costs associated with state A (VL undetected) | $166 | 1.0 | 3.50% | 3.50% | $324 | 1.0 | 3.50% | 3.50% |
| Direct medical costs associated with state B (VL<1000 copies/mL) | $166 | 0.6 | 3.50% | 3.50% | $324 | 0.6 | 3.50% | 3.50% |
| Direct medical costs associated with state C (VL>1000 copies/mL) | $256 | 0.3 | 3.50% | 3.50% | $449 | 0.3 | 3.50% | 3.50% |
| Direct medical costs associated with state D (Death) | $0 | – | – | – | $0 | – | – | – |

State A - VL_un-detected, state B - VL<1000 copies/mL), state C - VL>1000 copies/mL, state D – Death.

and the higher CATS to beneficiary supervision ratio (1:20 as a minimum) in implementation from 1:8 in the trial can be expected to lower programme costs and increase cost-effectiveness. We therefore varied the key evidence informing parameters (programme costs, health-related utilities, and the discount rate) to assess their influence on cost-effectiveness. The range of sensitivity analyses undertaken included: Cost per client year on ART and cost per client virally suppressed; +/-10%; Discount rate (uniform) for both costs (4% and 3%) and effects (3% and 4%); Utilities (QALYs) between 7.5 and 9.5 around the base case 0.85 observed in the Zvandiri Trial [51–54]. The best and worst-case scenarios, the point where all the parameters together yield the lowest/highest ICER, are presented.

## Ethics statement

Ethics approval for the Zvandiri Trial in which this cost effectiveness modelling is nested was granted by the Medical Research Council of Zimbabwe (A/2032). The Zvandiri trial was registered with the Pan African Clinical Trial Registry (PACTR201609001767322).

## Results

### Effect of the intervention after 40 cycles (40 years)

Our cohort-simulation model suggests that, under the status quo, and after 40-years (year 2058), of the cohort of 280 ALHIV clients, 59 (21%) will have undetectable VL, 33 (12%) detectable VL<1000 copies/mL, 27 (10%) VL≥1000 copies/mL, and 160 (57%) will have died (Table 3).

**Table 3. Effect of the intervention after 40 cycles (40 years).**

| State | HIV viral load undetected | VL < 1000 copies | VL > 1000 copies | Death |
|---|---|---|---|---|
| SoC | 21% (59/280) | 12% (33/280) | 10% (27/280) | 57% (160/280) |
| Zvandiri | 65% (183/280) | 23% (65/280) | 3% (8/280) | 9% (24/280) |

With additional implementation of the Zvandiri intervention over 40 years, adherence to ART improves for the duration of the intervention, decreasing the yearly probability of virological failure or death (Fig 3); after 40 years 183 (65%) individuals will have an undetectable viral load, 65 (23%) will have detectable VL < 1000/ copies/mL, 8 (3%) will have VL ≥ 1000 copies/mL and 24 (9%) will have died (Table 3).

### Cost-effectiveness

Table 4 presents deterministic cost-effectiveness results for the two arms, lifetime cost, LY and QALY outcomes as well as ICER results from the Markov model. SoC had a total programme cost of $822,553 over 40 years vs $1,323,140 for Zvandiri.

### Cost per LY and QALY gained

Based on the modelled parameters and costs, SoC results in 4,442 LYs over the 40 years and is "dominated" by Zvandiri which has higher LYs (5,787). Zvandiri results in 1,345 LYs gained at an incremental cost of $500,587 yielding a discounted ICER of $372 per LY gained. SOC (dominated) also results in 3,731 QALYs over the 40 years compared to Zvandiri, which has higher QALYs (4,977). Zvandiri results in 1,246 QALYs gained at an incremental cost of $123,645 yielding a discounted ICER of $99 per QALY gained (Table 4).

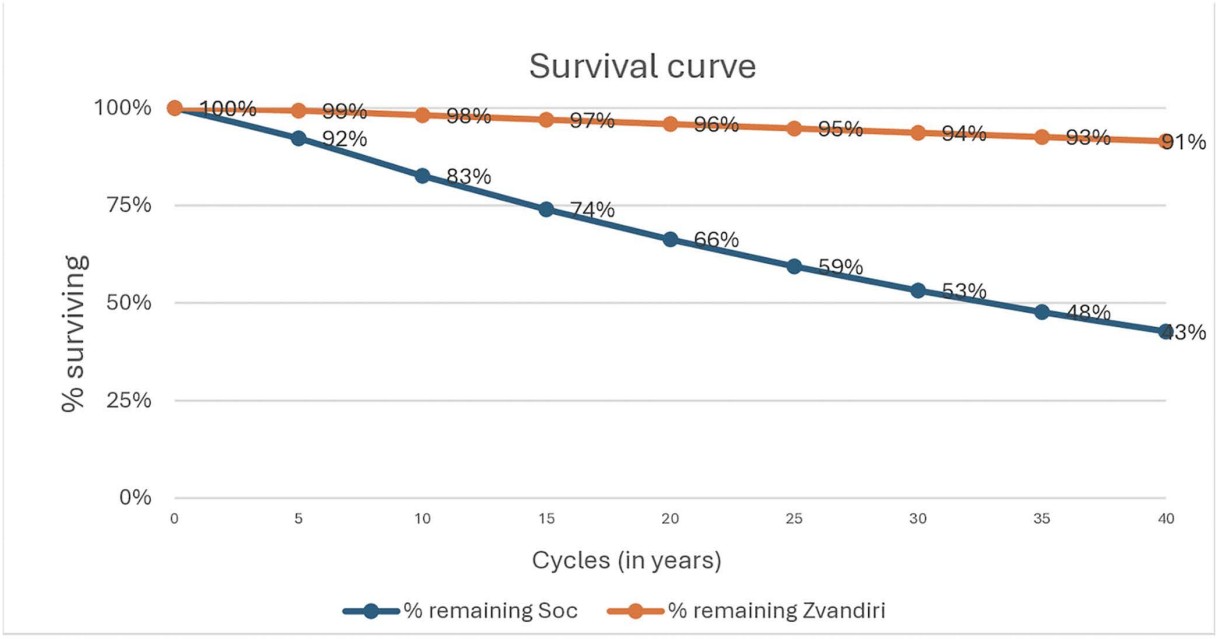

**Fig 3. Survival curve showing the expected duration of time until death in each cohort.**

**Table 4. Cost-effectiveness modelling outputs (LYs and QALYs).**

| | Standard of Care | | Zvandiri | | Difference | | Estimated ICER |
|---|---|---|---|---|---|---|---|
| | Life Years | Cost | Life Years | Cost | Life Years Gained | Incremental Costs | |
| Deterministic* Results-LYs gained | 4,442 | $822,553 | 5,787 | $1,323,140 | 1,345 | $500,587 | $372 |
| | QALYs | Cost | QALYs | Cost | QALY Gained | Incremental Costs | Estimated ICER |
| Deterministic* Results-QALYs gained | 3,731 | $822,553 | 4,977 | $946,198 | 1,246 | $123,645 | $99 |

*Deterministic analysis (unlike probabilistic analysis) evaluates the model using only parameter means of model inputs (point estimates of costs and outcomes) and gives only a single output for decision making. SoC is dominated by Zvandiri given lower effectiveness (LYs and QALYs gained). Probabilistic analysis evaluates the model over a distribution of these parameters and bases decisions on the distribution of outputs.

## Sensitivity analyses results

Fig 4 depicts results from one and multiway sensitivity analyses where we varied model input parameters to assess their impact on Zvandiri cost-effectiveness (ICER). The bars in the Tornado diagram represent changes in the ICER for each input parameter, from the base case. Wider bars represent parameters that contribute more to the variability of the ICER. The ICER was highly sensitive to the following evidence informing parameter estimates: programme economic costs, health-related utilities and, the discount rate (base case 3.5%). Varying cost per client year on ART and cost per client virally suppressed (+/-10%) yielded costs of $23.31 and $175.22. Constant discounting for programme costs between 3% and 4% resulted in costs of $44.49 and $160.88, while for effects (3% and 4%), it led to costs of $75.01 and $139.28. When varying utilities (QALYs) between 7.5 and 9.5, ICERs were $69.99 and $203, respectively. The best- and worst-case scenarios, the point where all the parameters together yield the lowest/highest ICER, were $15.91 and $1,156.11.

Modelling out different time horizons over which the intervention could be modelled out (5, 10, 15 and lifetime horizons) results in decreasing ICERs (and therefore, increasing cost-effectiveness) with time, as the number of life years gained each year increases over time (Fig 5).

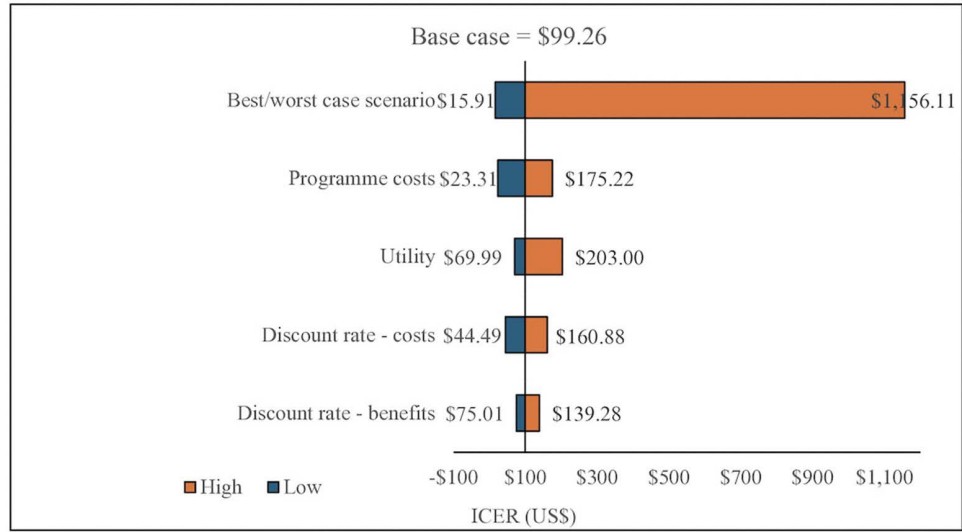

**Fig 4. Tornado diagram of one-way sensitivity analysis.**

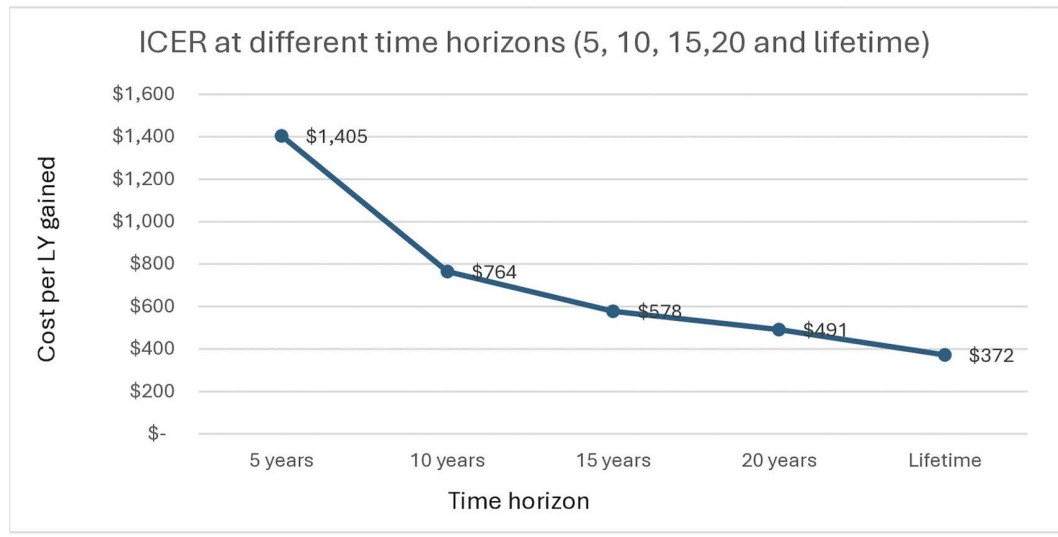

**Fig 5. ICER at different time horizons (5, 10, 15, 20 and lifetime).**

## Discussion

This modelling study is one of the first to examine the longer-term costs and economic consequences of community-based peer-delivered differentiated support for ALHIV in sub-Saharan Africa. Results from the Markov model suggest that over the long term, adherence to ART continues to improve, decreasing the annual probability of virological failure or death compared to SoC. Zvandiri has a higher proportion of individuals with undetectable viral load, and within the suppression threshold (detectable VL < 1000 copies/mL) compared to SoC. Zvandiri also results in higher LYs and QALYs gained compared to SoC.

The ICERs, $372.14 per LY or $99.26 per QALY gained, are below the commonly accepted ICER thresholds for LMICs [58–60]. Compared to the median cost-effectiveness threshold per LY or QALY of <$700 (representing a proportion of 0.53 of the country GDP) for LMICs, Zvandiri is highly cost-effective and remains so even under a more restricted <$500 per LY or QALY gained threshold, now considered appropriate for LMICs, but which most high prevalence countries would still struggle to afford [58–60].

Our cost-effectiveness results were robust to variations in input parameters. Varying single parameters in one way sensitivity analysis never yielded ICERs for Zvandiri exceeding even the lower <$500 LMIC threshold. Only in the best- and worst-case scenario analysis did the ICER breach the <$500 LMIC ($1,156) and the higher US$700 thresholds. Zvandiri is likely even more cost-effective in real-world implementation due to a number of cost reductions. Firstly, it has reduced human resources costs compared to those incurred during the trial by shifting supervision costs from more expensive NGO to MoH staff. Supervision of CATS is now undertaken by more experienced CATS employed directly by the National AIDS Council. In addition, the CATS to beneficiary supervision ratio has increased from 1:8 in the trial to 1:20 (as a minimum), which likely results in lower programme costs and an even more cost-effective ICER.

Our analysis is largely consistent with the current evidence on cost-effectiveness of comprehensive community-based health systems strengthening programmes to improve retention in care. Retention-in-care strategies compared to alternative resource uses for HIV-infected persons in East Africa were found to have potential for relatively inexpensive scaleup and are, therefore, a valuable next option on the implementation "menu" for policymakers and decision makers [12]. A study in Uganda assessing the efficacy and cost-effectiveness of a savings-led family-based economic empowerment

intervention, Suubi+Adherence, found incremental costs of virally suppressing one additional adolescent at US$970 did not seem prohibitive compared to other community-based adherence interventions targeted at ALHIV in low-resource settings [62,63].

In South Africa, community-based support provided by lay health workers for adolescents and youth receiving ART was associated with substantially reduced patient attrition, and was a low-cost intervention with reasonable cost-effectiveness that can aid progress towards several health, economic and equality-related sustainable development goals targets [64]. Another study in South Africa found targeted, task-shifted, cognitive behavioural therapy for people living with HIV with diagnosed depression and virological failure in a randomised control trial improved life expectancy, was cost-effective and should be integrated into HIV care [65]. Crucially, none of these studies took a long-term horizon as their objectives targeted short term cost-effectiveness. The results of our analyses not only extend our earlier CRT-based cost analysis and add to the above growing literature, but also provide new evidence on the longer-term (40 years) cost-effectiveness of community-based health systems strengthening programmes that improve retention in care.

## Strengths and limitations

As discussed earlier, though hybrid economic evaluations such as this one inform the modelling effort using truncated trial follow-up data, they address this limitation by projecting the results through time, generating a range of plausible projections of longer-term outcomes [38,39]. In this analysis, we used trial outcomes from a rigorous CRT conducted over two years to compute the state transition probabilities which informed assignment and tracking of patient movements over the life of the model (40 years). Model cost inputs were derived from a primary costing conducted alongside the CRT (prospective) [8]. The primary costing exercise involved facility visits or site observations to understand ART provision and community adherence support activities as well as time and motion analysis to track and best estimate allocation of provider time to ART relative to other clinical services [8].

We also used a 40-year time horizon to capture the lifetime costs and outcomes of ALHIV on ART, based on a Zimbabwe-specific life expectancy [41]. Cost-effectiveness analyses elsewhere have been criticised for having shorter time horizons, which do not allow effects to play out over a longer period.

In this study design, intervention efficacy was also assessed using data collected in the context of the CRT, minimising the selection and allocation bias risks associated with many past economic evaluations [63]. In addition, an objective and reliable indicator of adherence was used to assess efficacy, rather than relying on self-reported measures. Lastly, we believe our results are more robust as the half-cycle correction method allows ALHIV patient events or transitions to occur randomly throughout each cycle rather than at the beginning or end, reducing over- or underestimation of outcomes and the associated costs [55].

Our analyses are subject to some limitations. The CRT was conducted in an era prior to the programmatic transition to tenofovir, lamivudine and dolutegravir (TLD) in LMICs. TLD is a relatively novel oral formulation that increases rates of virological suppression. It is possible that ALHIV in the SoC arm will have higher rates of suppression than predicted here, but long term TLD effectiveness still relies on sustained adherence. This is particularly relevant due to the well-documented adherence challenges among ALHIV.

In practice, our model may underestimate cost-effectiveness by failing to account for all the benefits that can potentially accrue to an intervention such as this one. Firstly, unlike in a transmission model, our Markov design was limited to modelling only the costs and impacts of adherence on the immediate suppression outcomes from the trial. A more comprehensive infectious disease model would have captured the indirect benefits (through affecting transmission) of better ART retention and adherence (reduced VL), better reflecting real-life scenarios of importance to the modelled outcomes [66,67]. However, and for the purposes of our analyses, the potential reductions in onward transmissions were not considered an objective.

Secondly, as highlighted earlier with other models that use Markov chains, our model is limited by the "memoryless property/Markov assumption", which assumes that cohort models only depend on the current health state at any given cycle. It therefore, cannot depend on the history prior to that cycle and only investigates a hypothetical homogeneous (no heterogeneity) cohort of individuals transitioning across health states, with a result precisely governed within a set of initial conditions and parameters [31,45,46]. For this reason, our model is unable to capture the changes with each cycle, for example: attrition, the likelihood of increased risk of dying with age, as well as the impacts of all-cause mortality.

Additionally, and as outlined above, programmatic transition to TLD has potentially reduced Zvandiri costs and benefits (with less side effects and better virological outcomes) since the CRT. The results of this analysis which depend on pre-TLD era parameters may therefore significantly undervalue the intervention's cost-effectiveness.

Another limitation stems from our assumption that all deaths were due to HIV (background mortality equalled zero) allowing treated individuals to benefit from the intervention for the full 40-year projection horizon, even if they could have died from other causes and potentially overestimating the benefits.

From an economic evaluation point of view, a further limitation of this study is that it does not fully account for all economic costs as it relies only on provider costs excluding those that are borne by patients [31,35,68]. International reference cases recommend taking a disaggregated societal perspective when conducting economic evaluations of public health programmes to capture relevant non-health cost and effects falling outside the health system budget [31,35,36]. A patient costing would have allowed us to estimate direct out of pocket health expenses including transport costs as well as productivity losses and household expenses [69–71]. Globally, out of pocket health expenses cause financial hardship (at worst impoverishment) having pushed +/-100 million people into extreme poverty (below the $3.20 poverty line) in 2015 alone and taking up 10–25% of their consumption or income [71–75].

Lastly, as previously discussed, Zvandiri is a multi-component, differentiated service-delivery model implemented as an integrated package. A key limitation of such complex interventions is the challenge of disentangling the effects of individual components, which reduces the ability to generalise findings across settings. This represents a broader limitation of economic evaluations of multicomponent interventions: while it may be feasible to estimate the cost of each component, identifying which specific element contributes most to effectiveness remains difficult. In our primary costing study, Zvandiri was analysed as a single, bundled intervention, which precluded assessment of the relative cost-effectiveness of its individual components.

## Policy implications

In order for Zimbabwe to fully attain UNAIDS 95-95-95 targets, scaling up an effective community-based support strategy such as Zvandiri is key. Zvandiri utilises peer support to promote context-relevant, individual resilience-building to directly improve the wellbeing of ALHIV and strengthen their proactive engagement with services across the HIV prevention and care cascades and which are not available via the primary care system alone. Zvandiri was cost-effective compared to SoC at retaining a higher number of ALHIV patients on ART and at viral suppression.

## Conclusions

In Eastern and Southern Africa (ESA), the epicentre of the HIV epidemic, while significant progress has been made in scaling up ART for ALHIV, adherence to, and retention on, treatment has proven to be a major challenge in this population, driven by high rates of poor mental health, leading to poor health outcomes. Effective, socially and economically contextual community-based peer-led counselling and-support programmes that aim to improve the wellbeing and longevity of ALHIV are critical for ESA.

WHO recommends differentiated, youth-led, and multicomponent interventions to support ART adherence and retention in care in order to meet UNAIDS 95-95-95 targets. Our cost-effectiveness findings add to the growing evidence base that community-based interventions improve ART adherence outcomes among ALHIV. Our modelling analyses suggest

that Zvandiri, which effectively reduces rates of virological failure among ALHIV in Zimbabwe is highly cost-effective. The Zvandiri intervention should continue to be scaled up nationally, regionally and in other SSA countries with similar contexts (now adopted/localised in 14 countries).

## Supporting information

**S1 File. Zvandiri Markov Model Parameters.**
(XLSM)

## Acknowledgments

We thank MoHCC for permission, access to health facilities and support to conduct this study. We extend our gratitude to management and staff of Zvandiri for their support and assistance to access both programme financial and M&E data. We also thank Paul Revill of University of York for invaluable technical and manuscript support.

## Author contributions

**Conceptualization:** Webster Mavhu, Nicola Willis, Sarah Bernays, Walter Mangezi, Tsitsi Apollo, Ricardo Araya, Helen A. Weiss, Frances M. Cowan, Hendramoorthy Maheswaran.

**Formal analysis:** Collin Mangenah, Webster Mavhu, Alfredo Palacios, Fern Terris-Prestholt, Frances M. Cowan, Hendramoorthy Maheswaran.

**Funding acquisition:** Webster Mavhu, Nicola Willis, Sarah Bernays, Frances M. Cowan.

**Investigation:** Collin Mangenah, Webster Mavhu, Nicola Willis, Juliet Mufuka, Sarah Bernays, Walter Mangezi, Tsitsi Apollo, Ricardo Araya, Helen A. Weiss, Fern Terris-Prestholt, Frances M. Cowan, Hendramoorthy Maheswaran.

**Methodology:** Collin Mangenah, Webster Mavhu, Nicola Willis, Alfredo Palacios, Fern Terris-Prestholt, Frances M. Cowan, Hendramoorthy Maheswaran.

**Project administration:** Webster Mavhu, Nicola Willis, Juliet Mufuka, Sarah Bernays, Walter Mangezi, Frances M. Cowan.

**Supervision:** Webster Mavhu, Nicola Willis, Sarah Bernays, Walter Mangezi, Tsitsi Apollo, Ricardo Araya, Helen A. Weiss, Alfredo Palacios, Fern Terris-Prestholt, Frances M. Cowan, Hendramoorthy Maheswaran.

**Validation:** Webster Mavhu.

**Writing – original draft:** Collin Mangenah.

**Writing – review & editing:** Collin Mangenah, Webster Mavhu, Nicola Willis, Juliet Mufuka, Sarah Bernays, Walter Mangezi, Tsitsi Apollo, Ricardo Araya, Helen A. Weiss, Alfredo Palacios, Fern Terris-Prestholt, Frances M. Cowan, Hendramoorthy Maheswaran.

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
