## [Decision Letter · Decision Letter 0]

14 Aug 2025

PGPH-D-25-01759

Cost-effectiveness of Zvandiri, a community-based support intervention to reduce virological failure in adolescents living with HIV in Zimbabwe: Results of a decision analytical model

Dear Dr. Collin Mangenah,

Thank you for submitting your manuscript to PLOS Global Public Health. After careful consideration, we feel that it has merit but does not fully meet PLOS Global Public Health’s publication criteria as it currently stands. Therefore, we invite you to submit a revised version of the manuscript that addresses the points raised during the review process.

We look forward to receiving your revised manuscript.

Kind regards,

Vishal Goyal

Academic Editor

Journal Requirements:

1. We have amended your Competing Interest statement to comply with journal style. We kindly ask that you double check the statement and let us know if anything is incorrect.

2. We do not publish any copyright or trademark symbols that usually accompany proprietary names, eg (R), (C), or TM (e.g. next to drug or reagent names). Please remove all instances of trademark/copyright symbols throughout the text, including ® on page 9.

3. We note that your Data Availability Statement is currently as follows: [All data pertinent to the cost effectiveness findings are included in the manuscript. Data on the cohort of adolescents living with HIV is held and available upon reasonable request via a non-author point of contact:

Institution: Centre for Sexual Health and HIV/AIDS Research (CeSHHAR)

Contact person: Data Manager, Jeffrey Dirawo (Jeffrey.dirawo@ceshhar.org)]

**Comments to the Author**

Reviewer #1: Fully recommend.

**Minor revisions required as highlighted in the attached document.**

Well articulated manuscript, easy to follow and understand the study outline and methods. The data is well presented and the analyses are properly explained with interesting implications of a novel model of a 40-year time horizon. The strength and limitations of the analyses are clearly outlined with a possibility of the intervention's cost effectiveness being underestimated since the study was undertaken pre-TLD (dolutegravir) without accounting for reductions in HIV transmissions from the high viral suppression observed.

Such studies are imperative for the future of SSA in light of the current HIV funding crisis.

Reviewer #2: 

This is a significant study addressing growing concerns for a delicate age group, especially with the anticipated increase in the number of ALHIV that require and will continue to require optimum care.

Comment 1: It was not very clear in the methods at which time point did the cohort for the model start simulations? Was it at the start of the study (baseline)? Or at the end of the 96 weeks of the trial period.

Comment 2: Why did the authors decide to use Markov modelling methods instead of the non-Markov methods, especially since in HIV infection, past biological and behavioural factors influence future outcomes? What advantages does this method have over the rest, especially to increase the external validity of the study?

Comment 3: Did the model consider viral resistance? Or immunological failure? Attrition? How could this influence the outcomes?

Comment 4: Zvandiri is a complex intervention community model. But the authors considered Zvandiri as one intervention in the ICER calculations. The limitation of complex interventions is the inability to assess the impact of individual interventions on the outcome. This makes generalisability difficult. Are the authors able to perform some ranking to control for dominant interventions? Or present the results based on ranking? Otherwise, the authors need to support their strong conclusions with more evidence.

---

## [Decision Letter · Decision Letter 1]

3 Nov 2025

PGPH-D-25-01759R1

Cost-effectiveness of Zvandiri, a community-based support intervention to reduce virological failure in adolescents living with HIV in Zimbabwe: Results of a decision analytical model

Dear Dr. Collin Mangenah,

Thank you for submitting your manuscript to PLOS Global Public Health. After careful consideration, we feel that it has merit but does not fully meet PLOS Global Public Health’s publication criteria as it currently stands. Therefore, we invite you to submit a revised version of the manuscript that addresses the points raised during the review process.

We look forward to receiving your revised manuscript.

Kind regards,

Vishal Goyal

Academic Editor

Journal Requirements:

1. Please note that PLOS Global Public Health has specific guidelines on code sharing for submissions in which author-generated code underpins the findings in the manuscript. In these cases, all author-generated code must be made available without restrictions upon publication of the work. Please review our guidelines at https://journals.plos.org/globalpublichealth/s/materials-and-software-sharing#loc-sharing-code and ensure that your code is shared in a way that follows best practice and facilitates reproducibility and reuse.

-DOI: 10.1016/j.jval.2015.09.2940

-DOI:10.1097/qad.0000000000003440

In your revision ensure you cite all your sources (including your own works), and quote or rephrase any duplicated text outside the methods section. Further consideration is dependent on these concerns being addressed.

Reviewers' comments:

**Comments to the Author**

Reviewer #1: The authors rebuttal letter states that they attended to the minor comments however, none of the corrections are reflected in the revised manuscript. The authors need to revise the manuscript accordingly.

Reviewer #2: The authors made a more clear explanation of the statistical methods and parts of the discussion. This improved the quality of the manuscript.

---

## [Decision Letter · Decision Letter 2]

11 Nov 2025

Cost-effectiveness of Zvandiri, a community-based support intervention to reduce virological failure in adolescents living with HIV in Zimbabwe: Results of a decision analytical model

PGPH-D-25-01759R2

Dear Collin Mangenah,

We are pleased to inform you that your manuscript 'Cost-effectiveness of Zvandiri, a community-based support intervention to reduce virological failure in adolescents living with HIV in Zimbabwe: Results of a decision analytical model' has been provisionally accepted for publication in PLOS Global Public Health.

Best regards,

Vishal Goyal

Academic Editor
